# FOXP3 and CXCR4-positive regulatory T cells in the tumor stroma as indicators of tumor immunity in the conjunctival squamous cell carcinoma microenvironment

Mizuki Tagami[1,2]*, Anna Kakehashi[3], Atsuko Katsuyama-Yoshikawa[2,4], Norihiko Misawa[1], Atsushi Sakai[1], Hideki Wanibuchi[3], Atsushi Azumi[2], Shigeru Honda[1]

1 Department of Ophthalmology and Visual Sciences, Graduate School of Medicine, Osaka City University, Osaka, Japan, 2 Ophthalmology Department and Eye Center, Kobe Kaisei Hospital, Kobe, Hyogo, Japan, 3 Department of Molecular Pathology, Graduate School of Medicine, Osaka City University, Osaka, Japan, 4 Division of Ophthalmology, Department of Surgery, Kobe University Graduate School of Medicine, Hyogo, Japan

* tagami.mizuki@med.osaka-cu.ac.jp

## Abstract

Conjunctival squamous cell carcinoma (SCC) is the most common ocular surface neoplasia. The purpose of this retrospective study was to examine the role of regulatory T cell (Treg) activity in tumor immunity and investigate the tumor microenvironment as a new treatment focus in conjunctival SCC. Cancer progression gene array and immunohistochemical analyses of FOXP3 as a Treg marker, CD8 as a tumor-infiltrating lymphocyte marker, and CXCR4 expression on activated Tregs were conducted in a series of 31 conjunctival SCC cases. The objective was to investigate the immunoreactive response in tumor cells and stromal cells in the cancer microenvironment. The stroma ratio in tumor cells was investigated by monitoring α-smooth muscle actine (SMA) expression between carcinoma in situ (Tis) and advanced carcinoma (Tadv) ($P$<0.01). No significant change in PD-L1 expression was observed in this study ($P$ = 0.15). Staining patterns of FOXP3, CD8, and CXCR4 were examined separately between tumor cells and stromal cells in SCC tumors. Differences in staining of FOXP3 in Tregs and CD8 in tumor-infiltrating lymphocytes in tumor stroma in the Tis group were observed compared with the Tadv group (each $P$<0.01). In addition, double immunostaining of CXCR4/FOXP3 was correlated with progression-free survival ($P$ = 0.049). Double immunostaining of CXCR4/FOXP3 correlated with American Joint Committee on Cancer T-stage, independent of age or Ki67 index ($P$<0.01). Our results show that FOXP3 and the CXCR4/FOXP3 axis are important pathologic and prognostic factors of ocular surface neoplasia, including SCC. The tumor microenvironment of conjunctival SCC should be considered in the future development of treatment options.

**Data Availability Statement:** All relevant data are within the manuscript and its Supporting information files.

**Funding:** The authors received no specific funding for this work.

**Competing interests:** The authors have declared that no competing interests exist.

# Introduction

Ocular surface squamous neoplasia (OSSN) includes several diseases, such as conjunctival pre-malignant dysplasia, carcinoma in situ (Tis), and advanced conjunctival squamous cell carcinoma (Tadv) [1]. The annual incidence of OSSN in the United Kingdom is 0.53 cases/million population/year (conjunctival intraepithelial neoplasia: 0.43 cases/million population/year; SCC: 0.08 cases/million population/year) [2, 3]. In the United States, the incidence of SCC is 5-fold higher among males and whites [4]. However, no large-scale epidemiologic studies have been performed in populations in the Far East. In addition, recent studies have reported associations between findings such as loss of retinoblastoma protein/P16/cyclin D1, strong p53 immunostaining, and ADAM3A overexpression and the molecular pathology of SCC [5–7]. In recent years, research on factors related to tumor immunity, including programmed cell death 1 (PD-1), programmed cell death ligand 1 (PD-L1), and cytotoxic T lymphocyte–associated protein 4 (CTLA-4) has advanced worldwide to encompass aspects ranging from basic research to clinical indications in head-neck SCC [8–11]. In skin SCC, the efficacy of Pembrolizumab and Cemiplimab and the potential for first-line treatment have been reported [12, 13]. Recent ophthalmologic-related research demonstrated that PD-L1 is expressed in almost half of conjunctival SCC cases and indicated the potential application of immune checkpoint blockade as a treatment strategy for conjunctival SCC [6, 14]. Other research indicates that a significant number of sebaceous carcinoma cases involve expression of PD-L1 at therapeutic levels with tumor-infiltrating T cells.

Regulatory T cells (Tregs) play a major role in tumor immunity in the tumor microenvironment [15–19]. In particular, CD4(+) Tregs expressing the transcription factor forkhead box P3 (FOXP3) are abundant in tumor tissues, where they appear to hinder the induction of effective antitumor immunity [15]. Tregs function in maintaining immunologic self-tolerance by actively suppressing self-reactive lymphocytes, and the gene encoding FOXP3 is a key regulator of Treg development [20]. Freeman et al. reported that a high CD4+ T cell to CD8+ T cell ratio may be an immunologic diagnostic indicator of late-stage dermal SCC development in immune-competent patients [21]. Considerable research attention has focused on the relationship between the role of local tumor-infiltrating lymphocytes (TILs [CD4+ and/or CD8+ T cells]) associated with FOXP3+ Tregs in the tumor microenvironment and the prognosis of SCC patients. Elucidating details regarding the interactions between Tregs in the tumor microenvironment and other molecules is another important area of research. The present study investigated the expression of FOXP3 in tumor tissues from SCC patients in order to assess the molecular association between Tregs and prognosis in East Asian patients with conjunctival SCC.

# Materials and methods

## Selection of cases and collation of clinicopathologic data

This study was approved by the Institutional Review Boards of Osaka City University and Kobe Kaisei Hospital and adhered to the tenets of the 1964 Declaration of Helsinki. Written informed consent was obtained from all patients before enrollment. We identified 29 patients treated by ophthalmologists (AA, MT) between November 2007 and April 2020 from whom we were able to procure tissue blocks with residual tumors. Demographic information (age at initial diagnosis and at presentation to our institution; sex) and primary tumor features (disease status at presentation [primary or recurrent] and in situ versus invasive disease) were obtained from each patient enrolled. The American Joint Committee on Cancer (AJCC) stage, local recurrence (anatomic site and date), metastases (regional or distant, and date), vital status

at last follow-up, cause of death if deceased, type of surgery, and adjuvant therapy were also recorded.

## Immunohistochemistry

Immunohistochemical analyses of FOXP3 and CD8 expression were performed on 3-μm-thick formalin-fixed paraffin-embedded (FFPE) tissue sections using the following antibodies: anti-human FOXP3 mouse monoclonal (clone: 236A/E7; #ab20034; Abcam, Cambridge, UK), anti-human CD8 mouse monoclonal (clone: 4B11; NCL-L-cd8-4B11; Leica, Newcastle, UK), anti-human CXCR4 rabbit polyclonal (NB100-74396; Novus, Continental, CO), anti-human α-smooth muscle actine (SMA) mouse monoclonal (clone: 1A4; 412021; Nichirei Bioscience, Tokyo, Japan), anti-human PD-L1 rabbit monoclonal (clone: 28–8; #ab205921; Abcam), anti-Ki67 rabbit monoclonal antibody (clone: SP6; ab16667; Abcam). Elite ABC Rabbit kit and Elite ABC Mouse kit (PK-6101, PK-6102; Vector Laboratories, Burlingame, CA), and ABC-AP Mouse IgG kit (AK-5002; Vector Laboratories). Tissue sections were incubated in ImmPACT DAB (Vector Laboratories) and Alkaline Phosphatase Substrate kit III <VECTOR Blue> (Vector Laboratories) until the desired staining intensity developed. The sections were then counterstained with hematoxylin and mounted; FOXP3/CXCR4 double staining was conducted without counterstain. Positive and negative staining controls for all antibodies were carried out in parallel using tonsillar tissue. Stained sections were viewed under an Olympus BX53+DP74 microscope.

## Image analysis

Tissues immunostained for FOXP3, CD8, α-SMA, PD-L1, and CXCR4 were evaluated in a blinded manner by two specialists (MT and AK). The first field for evaluation in each tumor lesion was selected randomly, and subsequently, 10 fields were examined systematically at 400× magnification using a mesh.

FOXP3 and CD8 expression was analyzed visually as the presence or absence and intensity of cell staining, and samples were divided semi-quantitatively into groups based on a score of 0 to 3 (0, none: 0–1 field; 1, weak: 1–5> 1 field; 2, strong: 5–10> 1 field; 3, very strong 10> 1 field).

α-SMA expression was analyzed in the tumor stroma for each group based on a score of 0 to 3 (stromal ratio: stroma/Whole tumor (%):0, none; 1, <30%; 2, 30–50%; 3, >50%).

PD-L1 expression was analyzed to determine the ratio of PD-L1 expression in different cells for all tumor groups based on a score of 0 to 2 (PD-L1 ratio: PD-L1 staining cells/whole tumor (%) (0, none:, <2%:1, 1–10%:2, 10%>:3).

In order to investigate localization in greater detail, the FOXP3 and CD8 staining patterns in tissues were examined separately for tumor (TM) and tumor stroma (ST). For FOXP3/CXCR4 double staining, the number of Tregs co-expressing FOXP3/CXCR4 was determined from four (×40) high-power fields of lymphoid-rich infiltrate within the tumor, according to previously reported methods [22]. Samples were semi-quantitatively divided into groups based on a FOXP3/CXCR4 double staining score in the range 0 to 3 (0, none: 0:1 field; 1, weak: 1–2> 1 field; 2, strong: 3> 1 field) according to the presence or absence and intensity of staining.

## Immunofluorescence

The FFPE sections (3 μm) were deparaffinized for fluorescence immunohistochemistry. For heat-induced antigen retrieval, tissue sections were immersed in 0.01 mol/L citrate buffer and treated in a microwave oven at 620 W. The sections were then blocked with 1% bovine serum albumin for 1 h at room temperature. Sections were then incubated with the primary antibody,

anti-FOXP3 (1:50) or anti-CXCR4 (1:50), overnight at 4˚C, washed with PBS, and incubated with donkey anti-rabbit-Alexa Fluor 488 (1:1000; Thermo Fisher Scientific, Tokyo, Japan), goat anti-mouse-Alexa Fluor 594 (1:1000, #ab150073, Abcam) for 30 min at room temperature and placed on slides using mounting medium with DAPI (Vectashield antifade medium with DAPI: Vector Laboratories).

## Gene expression in tumors

The expression of major cancer-associated genes in tumor tissues was compared between patients with Tis and Tadv using NanoString analysis.

Genes examined included *EGFR*, *HIF1A*, *ICAM1*, *IL-6*, *CXCR-4*, *MAPK1*, *MMP2*, *MMP9*, *NFKB1*, *NOTCH1*, *ROCK1*, *ROCK2*, *TGFB1*, *TNF*, *VEGFA*, and *WNT5A*. Archival formalin-fixed paraffin-embedded tumor tissues were retrieved and manually macrodissected. Total mRNA was isolated using a Qiagen miRNeasy kit (Qiagen, Valencia, CA, USA) according to the manufacturer's instructions. Isolated RNA was quantified using a NanoDrop system (Thermo Scientific, Wilmington, DE, USA) and regarded as adequate if a sample contained a minimum of 400 ng. The samples were subsequently analyzed using an nCounter PanCancer Progression Panel (NanoString, Seattle, WA, USA) according to the manufacturer's instructions. NanoString data were processed using the R statistical programming environment (v3.4.2). Considering counts obtained for positive control probe sets, raw NanoString counts for each gene were subjected to technical factorial normalization, which was carried out by subtracting the mean count plus two standard deviations from the CodeSet inherent negative controls. Subsequently, biological normalization using the included mRNA reference genes was performed. Additionally, all counts with $P>0.05$ by one-sided $t$-test versus negative controls plus two standard deviations were interpreted as not expressed over basal noise.

## Statistical analyses

Clinical and histopathologic characteristics were summarized using descriptive statistics. Correlations between immunohistochemical, demographic, and clinicopathologic factor data were assessed using the Wilcoxon rank sum and Fisher's exact tests. Analysis of covariance was used to analyze the relationship between T-stage of tumors and FOXP3/CXCR4-positive Tregs. With respect to regression analyses, an explanatory variable that roughly divided the number of cases by 15 was considered appropriate [23]. Progression-free survival (PFS) was defined as the time from surgery to disease recurrence or death from any cause. Cox regression modeling was used to evaluate correlations between clinicopathologic and immunohistochemical features and survival outcomes. Statistical analyses were performed using SPSS Statistics software, version 22 (IBM Japan, Tokyo, Japan). $P<0.05$ was considered indicative of statistical significance.

## Results

Clinicopathologic findings of our cohort are summarized in Table 1. All 31 patients (100%) were East Asian and included 17 men and 14 women, with a mean age at presentation of 77.9 years. Sixteen patients (51%) had invasive SCC, and 15 (49%) had an in situ tumor. Primary orbital exenteration was necessary for local disease control in three patients (9%), and two patients (6%) underwent additional orbital exenteration. Nine patients (29%) underwent adjuvant therapy, most commonly additional local surgery. Topical chemotherapy and radiation therapy were performed in one patient in the adjuvant therapy group. One patient in the adjuvant therapy group died with disease 11 months after diagnosis of regional and lung metastases; another patient was alive without disease at 50 months after diagnosis of regional

**Table 1. Clinicopathologic findings of 31 cases of conjunctival squamous cell carcinoma.**

|  | All (N = 31) |
| --- | --- |
|  | n (%) |
| **Age, years** |  |
| Mean (range) | 77.9 (63–98) |
| **Sex** |  |
| Male | 17 (54) |
| Female | 14 (46) |
| **Follow-up duration after primary Surgery Months (range)** | 28.2 (6–135) |
| **T-stage (AJCC)** |  |
| Tis | 15 (49) |
| T1 | 4 (12) |
| T2 | 3 (10) |
| T3 | 7 (23) |
| T4 | 2 (6) |
| **Primary surgery type** |  |
| Local excision | 28 |
| Orbital exenteration | 3 |
| **Adjuvant therapy** |  |
| No | 22 (70) |
| Yes | 9 (30) |
| Additional excision | 7 |
| Topical chemotherapy | 1 |
| Radiation therapy | 1 |
| **Immunohistochemical markers** |  |
| **Ki67 labeling index [22]** |  |
| ≥50% | 7 (22) |
| <50% | 24 (78) |
| Outcome |  |
| **Orbital exenteration** |  |
| Yes | 5 (16) |
| No | 26 (84) |
| **Local recurrence after curative therapy** |  |
| Yes | 7 (22) |
| No | 24 (78) |
| **Metastasis** |  |
| Distant | 0 (0) |
| **Regional + Distant** | 2 (6) |
| Regional | 1 (3) |
| None | 27 (91) |
| **Vital status at last follow-up** |  |
| Deceased | 3 (10) |
| Alive | 28 (90) |
| **Cause of death** |  |
| **Conjunctival SCC (metastasis)** | 2 (75) |
| Other | 1 (25) |

**Table 2. FOXP3 and CD8 staining patterns in the tumor and tumor stroma.**

|  | FOXP3 Tumor | FOXP3 Tumor stroma | CD8 Tumor | CD8 Tumor stroma |
|---|---|---|---|---|
| **Tis** **n = 15** | 0.46±0.56 | 1.0±0.93 | 0.86±0.69 | 1.26±1.28 |
| **Tadv** **n = 16** | 2.13±2.07 | 2.60±2.57 | 0.60±0.64 | 2.0±0.40 |
| ***P*** | <0.01* | <0.01* | 0.35 | <0.01* |

Tis, carcinoma in situ; Tadv, advanced carcinoma.

*Un-paired *t*-test.

metastases. Three patients (9%) died, two due to conjunctival SCC (described above). Nine patients (31%) experienced local recurrence after curative surgery (Table 1).

Scores for FOXP3 TM staining were 0.46±0.56 in the Tis group and 2.13±2.07 in the advanced carcinoma Tadv group (*P*<0.01). Scores for FOXP3 ST staining were 1.0±0.93 in the Tis group and 2.60±2.57 in the Tadv group (*P*<0.01) (Table 2). Scores for CD8 TM staining were 0.86±0.69 in the Tis group and 0.60±0.64 in the Tadv group (*P* = 0.35). Scores for CD8 ST staining were 1.26±1.28 in the Tis group and 2.0±0.40 in the Tadv group (*P*<0.01) (Fig 1 and Table 2).

In the evaluation of the stroma ratio in tumor tissue based on α-SMA expression, scores were 1.13±0.35 in the Tis group and 1.81±0.83 in the Tadv group (*P*<0.01) (Fig 2). There was no significant change in PD-L1 expression in our study (*P* = 0.15).

Expression of *EGFR*, *HIF1A*, *ICAM1*, *IL-6*, *CXCR4*, *MAPK1*, *MMP2*, *MMP9*, *NFKB1*, *NOTCH1*, *ROCK1*, *ROCK2*, *TGFB1*, *TNF*, *VEGFA*, and *WNT5A* was compared between the

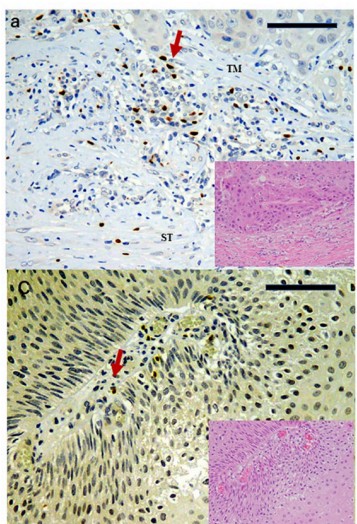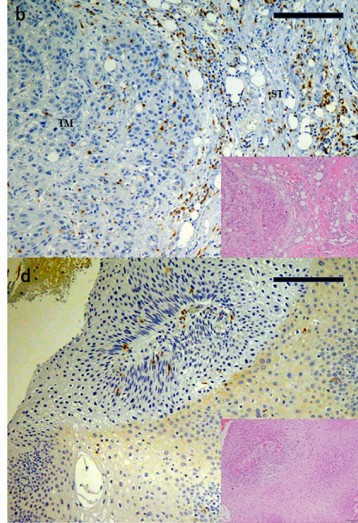

**Fig 1. FOXP3 and CD8 expression in conjunctival SCC tumor tissue.** (a) Strong FOXP3 staining in tumor and tumor stroma in invasive case(score: 3) (scale bar: 50 μm). (b) Strong CD8 staining in the tumor stroma in invasive case (score: 3) (scale bar: 100 μm). (c) Weak FOXP3 staining in carcinoma in situ case(score: 1) (scale bar: 50 μm). (d) Weak CD8 staining in tumor stroma in carcinoma in situ case (score: 3) (scale bar: 100 μm). Red arrow in (b) denotes FOXP3-positive lymphocyte. TM: Tumor; ST: Tumor stroma.

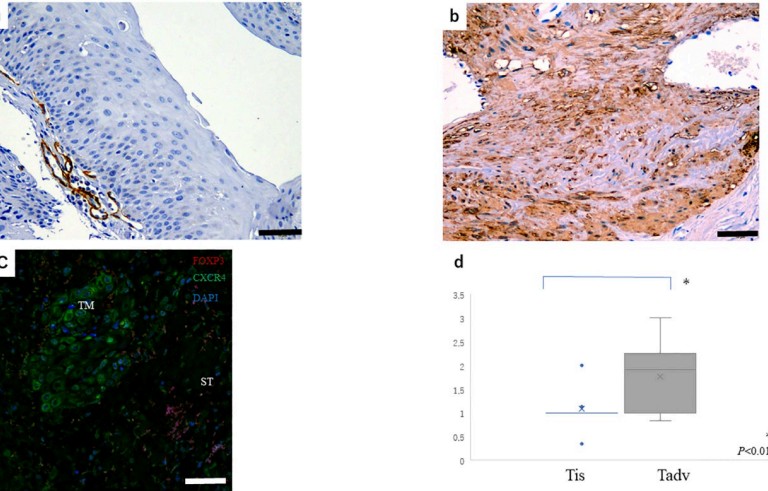

**Fig 2. α-SMA expression for stromal scoring in tumor tissue.** (a) Score: 1 (scale bar: 100 μm). (b) Score: 3 (scale bar: 100 μm). (c) Tumor (TM) and tumor stroma (ST) were examined by immunofluorescence using specific antibodies (FOXP3, CXCR4, and DAPI). Scale bar: 20 μm. (d) A higher proportion of Tadv cancers expressed a high stromal score (score: 1.81) compared with Tis cancers (score: 1.12) (*P*<0.01).

Tis and Tadv groups according to AJCC T grading (n = 4 in each group) (Table 3). Expression of mRNA for *CXCR4*, *HIF-1α*, *IL-6*, *NFKB1*, and *ROCK1* differed significantly (*P*<0.05) between the Tis and Tadv groups. We have already published some of these results [24]. the multiple correction test was also performed by the Benjamini-Yekutieli method, however in that case were no significant difference (P <0.05).

**Table 3. Gene expression–associated cancer progression between Tis and Tadv.**

|  | Log2 fold-change | Std error (log2) | Lower confidence limit (log2) | Upper confidence limit (log2) | *P*-value | probe ID |
|---|---|---|---|---|---|---|
| **CXCR4-mRNA** [22] | <u>**3.93**</u> | 0.668 | 2.62 | 5.24 | <0.01* | NM_003467.2:1335 |
| **EGFR-mRNA** | −0.51 | 0.32 | −1.14 | 0.116 | 0.162 | NM_201282.1:360 |
| **HIF1A-mRNA** [22] | <u>1.62</u> | 0.413 | 0.813 | 2.43 | <0.01* | NM_001530.2:1985 |
| **ICAM1-mRNA** | 1.11 | 0.693 | −0.248 | 2.47 | 0.16 | NM_000201.2:2253 |
| **IL6-mRNA** [22] | <u>**4.03**</u> | 1.1 | 1.88 | 6.19 | 0.0145* | NM_000600.1:220 |
| **MAPK1-mRNA** | −0.84 | 0.472 | −1.76 | 0.0856 | 0.126 | NM_138957.2:430 |
| **MMP2-mRNA** | 1.89 | 0.939 | 0.044 | 3.73 | 0.0916 | NM_004530.2:2360 |
| **MMP9-mRNA** | 2.85 | 1.17 | 0.549 | 5.15 | 0.0513 | NM_004994.2:1530 |
| **NFKB1-mRNA** [22] | <u>−2.32</u> | 0.344 | −3 | −1.65 | <0.01* | NM_003998.2:1675 |
| **NOTCH1-mRNA** | 0.219 | 0.493 | −0.746 | 1.18 | 0.672 | NM_017617.3:735 |
| **ROCK1-mRNA** [22] | <u>0.731</u> | 0.273 | 0.196 | 1.27 | 0.0367* | NM_005406.1:2660 |
| **STAT3-mRNA** | 0.834 | 0.385 | 0.0799 | 1.59 | 0.0733 | NM_139276.2:4535 |
| **ROCK2-mRNA** | 0.503 | 0.264 | −0.0152 | 1.02 | 0.106 | NM_004850.3:3140 |
| **TGFBI-mRNA** | −0.495 | 0.977 | −2.41 | 1.42 | 0.631 | NM_000358.2:2030 |
| **TNFSF10-mRNA** | 0.00102 | 0.724 | −1.42 | 1.42 | 0.999 | NM_003810.2:115 |
| **VEGFA-mRNA** | −0.375 | 0.731 | −1.81 | 1.06 | 0.626 | NM_001025366.1:1325 |
| **WNT5A-mRNA** | −1.68 | 0.697 | −3.04 | −0.31 | 0.0529 | NM_003392.3:475 |

Tis, carcinoma in situ (n = 4); Tadv, advanced carcinoma (n = 4).

*Un-paired *t*-test (*P*<0.05). Statistically significant differences are underlined.

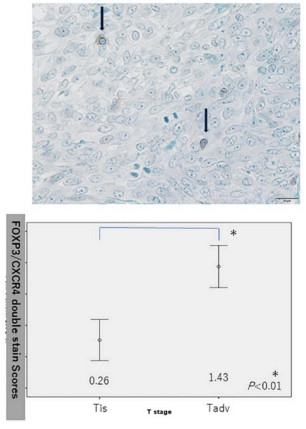
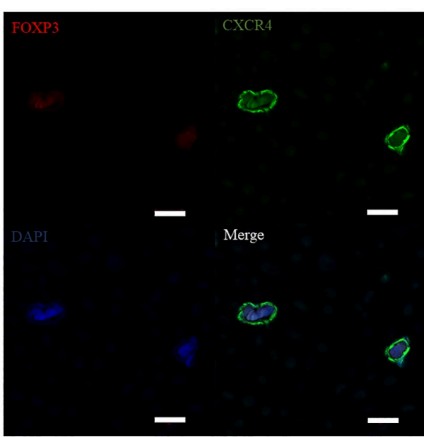

**Fig 3. Double CXCR4/FOXP3 immunostaining to evaluate CXCR4-positive Treg infiltration in whole tissue samples.** (Left upper) Black arrows: double staining, nuclear (blue: FOXP3), and cytoplasm and cell membrane (brown: CXCR4) (scale bar: 20 μm). (Left lower) A higher proportion of Tadv cancers expressed CXCR (score: 1.43) compared with Tis cancers (score: 0.26) (*P*<0.01). (Right) Double CXCR4/FOXP3 staining in Tregs was examined by immunofluorescence using specific antibodies. Scale bar = 20 μm.

Double CXCR4/FOXP3 immunostaining was performed to evaluate CXCR4-positive Treg infiltration in whole samples. A higher proportion of Tadv tumors expressed CXCR (score: 1.68) compared with Tis tumors (score: 0.13) (*P*<0.01) (Fig 3). Analysis of covariance indicated that double CXCR4/FOXP3 immunostaining was also correlated with T-stage, independent of age and Ki67 index (*P*<0.01).

The Cox regression model was used to examine the relationship between long-term prognosis (including orbital exenteration and PFS) and clinicopathologic status, FOXP3 and CD8 staining pattern, and double CXCR4/FOXP3 staining. Univariate Cox regression analyses revealed significant correlations between double CXCR4/FOXP3 staining and PFS (hazard ratio [HR]: 3.11; *P* = 0.032) (Table 4). Local recurrence, distant metastasis rate, and overall survival rate were not significantly correlated. In addition, FOXP3 and CD8 staining pattern and double CXCR4/FOXP3 staining were not significantly correlated with final orbital exenteration.

**Table 4. Relationship between progression-free survival and various clinicopathologic and molecular factors.**

| | Univariate analysis | | | |
|---|---|---|---|---|
| **Variable** | **N = 31** | **HR** | **95% CI** | ***P*** |
| **Age** | Mean 77.9 years | 1.305 | 0.904–1.882 | 0.155 |
| **Sex** | Male 17, Female 14 | 3.294 | 0.339–32.035 | 0.304 |
| **T-stage (AJCC)** | Tis: 15, >T1: 16 | 3.428 | 0.356–33.031 | 0.287 |
| **Ki67 index (%)** | Average: 30.58% | 1.017 | 0.987–1.049 | 0.266 |
| **Double CXCR4/FOXP3** | Tis: 0.13 T >1: 1.68 | **3.112** | 1.003–8.782 | 0.032* |

AJCC, American Joint Committee on Cancer; CI, confidence interval; HR, hazard ratio.

Statistically significant association is underlined.

*Cox proportional hazard model.

## Discussion

To the best of our knowledge, this is one of the first studies to investigate immunity in the tumor microenvironment in conjunctival SCC and evaluate the prognostic significance of TILs expressing FOXP3, CD8, and CXCR4/FOXP3 comparing the Tis and Tadv groups.

In addition, evaluation of the stroma using α-SMA and the expression of PD-L1 were also investigated in SCC and have been previously reported in other regions [25, 26].

In this study, we found a clear difference in expression of FOXP3 between the Tis and Tadv groups in TM and ST. Expression of CD8 was also significantly different in the ST group. Stromal ratios in tumor cells based on α-SMA expression differed significantly between the Tis and Tadv groups. α-SMA expression was high in the stroma of tumor tissue of patients with advanced stage. However, there was no significant difference in PD-L1 expression in our study.

In addition, CXCR4/FOXP3 double staining in Treg was carried out to confirm the TIL balance in the tumor microenvironment. There was a significant difference between the Tis and Tadv groups. Increased expression of CXCR4 in the Tadv group was also confirmed by both immunohistochemical analysis and our previous analysis of mRNA expression [24].

Recent research has shown that immune dysregulation, polyendocrinopathy, enteropathy, and X-linked syndrome are caused by mutations in *FOXP3*, indicating that this is the master gene of Tregs [20]. The primary mediator cells in the tumor microenvironment are Tregs, and many studies have investigated FOXP3 expression and the CD8 or other lymphocyte marker ratios in TILs as Treg markers [15, 22]. In ophthalmologic research, some studies have reported that Tregs play a role in protecting tissues from autoimmune diseases, such as dry eye or graft versus host disease after bone marrow transplantation, as well as ocular malignant neoplasms [27, 28]. However, some studies in the field of systemic oncology have reported that Tregs block TILs in the tumor microenvironment, thus promoting tumor progression. Therefore, local suppression of Tregs in the tumor is considered a new axis of targeted treatment, and a number of studies examining this possibility are underway [17–19].

Our study is the first to examine the expression of FOXP3, CD8, and CXCR4/FOXP3 in conjunction with tumor progression and prognosis in patients with conjunctival SCC. This is a new finding regarding tumor escape from immunity, which was also shown in our case of conjunctival SCC. In addition, double immunostaining of CXCR4 and FOXP3 in conjunctival SCC indicated that these factors may play cooperative molecular roles. When a tumor progresses, the ratio of FOXP3 to CD8 expression shifts toward FOXP3. This is consistent with previous studies of SCC tumors in other areas and suggests that the change in the FOXP3/CD8 ratio might affect tumor growth by enabling tumor cells to evade attack by TILs in the tumor microenvironment [8–10]. In addition. our results also show that double staining of CXCR4 / FOXP3 is significantly associated with PFS in the COX hazard model. However, due to the low number of cases and the low incidence of events, multivariate analysis has not been performed and confounding factors cannot be ruled out.

In order to investigate the mechanism by which tumor cells induce Tregs, we comprehensively examined the expression of genes thought to play a major role in tumor growth. We found that inflammation- and ischemia-related factors such as IL-6 and HIF-1α were overexpressed. Previous studies have reported that these factors induce Tregs to proliferate and aggregate into tumors [29–31]. Furthermore, vascular endothelial growth factor (VEGF)-associated ischemia has been shown to inhibit dendritic cell maturation via the NF-κB pathway, which would diminish the presentation of cancer cell antigens and promote the growth of cancer cells in the ocular microenvironment [32]. Indeed, in our study, expression of NF-κB was significantly lower in the Tadv group than the Tis group, although this was evaluated only at the mRNA level.

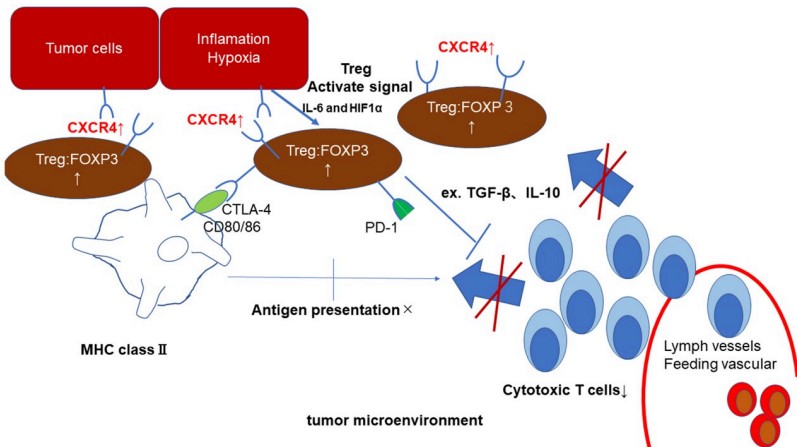

**Fig 4. Schematic illustration of hypothetical tumor immunity process in conjunctival squamous cell carcinoma.**

Among the genes examined, we were very interested in the expression of *CXCR4*, as CXCR4 is reportedly associated with tumor growth, and some studies indicated that CXCR4 affects the proliferation and activity of Tregs in the tumor microenvironment [22, 33, 34]. In addition, a relationship between CXCR4 and FOXP3-positive Tregs has been reported [22]. The results of our study also suggest that IL-6 and HIF-1α are associated with CXCR4 and that CXCR4 expression may be associated with an increase in FOXP3-positive Tregs. This means that inflammation is the anti-tumor immune effect of CXCR4-axis Tregs due to ischemia, and to our knowledge, ours is the first report of this association in the area of conjunctival SCC (Fig 4).

These findings have important implications in SCC, which is strongly resistant to targeted therapies but could potentially respond to immunotherapies that target Tregs. Furthermore, it is possible that Treg recruitment by CXCR4 in these cancers could be modulated by treatment directed against hypoxia pathway factors, including HIF-1α (Fig 4). Thus, the development of treatment strategies based on reagents directed against Tregs, antibodies that block CXCR4, or inhibitors of HIF-1α and VEGF represents a potentially fruitful area of research.

This study has important limitations. First, regarding FOXP3 and CXCR4 expression in OSSN, changes associated with benign disease and age-related changes in normal tissues may not have been sufficiently investigated. Further studies, including multi-institutional studies and an increase in the number of cases, will be needed in the future.

In gene expression array, the methods was macro-dissection, therefore, in some cases, there might be no distinction between tumor tissue and stroma. For that reason, this difference in genetic expression may be a combination of the tumor and the stroma around the tumor.

In addition, the size of our study cohort was small (N = 31), and the length of follow-up (less than 1 year in some patients) may not have been sufficient for long-term outcome analyses. Therefore, additional studies will be needed to corroborate our findings.

In conclusion, the results of this study indicate that FOXP3 and CXCR4, which are indicators of Treg activity, are potential molecular targets and prognostic factors in the treatment of OSSNs, including SCC. Our findings also suggest that the tumor microenvironment of conjunctival SCC must be considered in the future development of treatment options.

## Supporting information

**S1 Data.**
(XLSX)

## Acknowledgments

We gratefully acknowledge the technical assistance of the Research Support Platform, Osaka City University Graduate School of Medicine, and the Clinical Laboratory Department of Kobe Kaisei Hospital.

## Ethics approval

All procedures performed in studies involving human participants were conducted in accordance with the ethical standards of the Institutional and/or National Research Committee and with the 1964 Declaration of Helsinki and its later amendments or comparable ethical standards. Approval for this study was obtained prior to the start of the study from the institutional review board at Osaka City University, Japan (IRB-4236).

## Author Contributions

**Conceptualization:** Mizuki Tagami.

**Data curation:** Anna Kakehashi, Atsuko Katsuyama-Yoshikawa, Norihiko Misawa, Atsushi Sakai.

**Formal analysis:** Mizuki Tagami, Anna Kakehashi, Atsuko Katsuyama-Yoshikawa, Norihiko Misawa, Atsushi Sakai.

**Funding acquisition:** Shigeru Honda.

**Investigation:** Mizuki Tagami, Norihiko Misawa, Atsushi Sakai.

**Methodology:** Norihiko Misawa.

**Project administration:** Mizuki Tagami.

**Supervision:** Anna Kakehashi, Hideki Wanibuchi, Atsushi Azumi, Shigeru Honda.

**Visualization:** Atsuko Katsuyama-Yoshikawa.

**Writing – original draft:** Mizuki Tagami.

**Writing – review & editing:** Hideki Wanibuchi, Atsushi Azumi, Shigeru Honda.

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
