## [Decision Letter · Decision Letter 0]

31 Jan 2022

FOXP3 and CXCR4-positive regulatory T cells in the tumor stroma as indicators of tumor immunity in the conjunctival squamous cell carcinoma microenvironment Short titile: Regulatory T cells in conjunctival squamous cell carcinoma

PONE-D-21-21699

Dear Dr. Tagami,

We’re pleased to inform you that your manuscript has been judged scientifically suitable for publication and will be formally accepted for publication once it meets all outstanding technical requirements.

Kind regards,

Afsheen Raza, PhD

Academic Editor

PLOS ONE

Additional Editor Comments (optional):

Reviewers' comments:

Reviewer's Responses to Questions

**Comments to the Author**

1. Is the manuscript technically sound, and do the data support the conclusions?

Reviewer #1: Yes

2. Has the statistical analysis been performed appropriately and rigorously? 

Reviewer #1: Yes

3. Have the authors made all data underlying the findings in their manuscript fully available?

Reviewer #1: Yes

4. Is the manuscript presented in an intelligible fashion and written in standard English?

Reviewer #1: Yes

5. Review Comments to the Author

Reviewer #1: The authors performed cancer progression gene array and immunohistochemical analyses of FOXP3 as a Treg marker, CD8 as a tumor-infiltrating lymphocyte marker, and CXCR4 expression on activated Tregs in a series of 31 conjunctival SCC cases in order to investigate the immunoreactive response in tumor cells and stromal cells in the cancer microenvironment. The stroma ratio in tumor cells was investigated by monitoring α- smooth muscle actine (SMA) expression between carcinoma in situ (Tis) and advanced carcinoma (Tadv) (P<0.01). The authors conclude that no significant change in PD-L1 expression was observed in this study (P=0.15).

The manuscript is well written and data presentation as well as discussion of the results are adequate.

6. PLOS authors have the option to publish the peer review history of their article (what does this mean?). If published, this will include your full peer review and any attached files.

Reviewer #1: **Yes: **Prof. Satya N Das

---

## [Editor Report · Acceptance letter]

22 Mar 2022

PONE-D-21-21699 

FOXP3 and CXCR4-positive regulatory T cells in the tumor stroma as indicators of tumor immunity in the conjunctival squamous cell carcinoma microenvironment 

Dear Dr. Tagami:

I'm pleased to inform you that your manuscript has been deemed suitable for publication in PLOS ONE. Congratulations! Your manuscript is now with our production department. 

Kind regards, 

on behalf of

Dr. Afsheen Raza 

Academic Editor

PLOS ONE